# Pneumatic equiaxial compression device for mechanical manipulation of epithelial cell packing and physiology

**Heidi Peussa**[ID], **Joose Kreutzer**[*], **Elina Mäntylä**[ID][*], **Antti-Juhana Mäki**[ID], **Soile Nymark**, **Pasi Kallio**[ID], **Teemu O. Ihalainen**[ID]*

Faculty of Medicine and Health Technology, Tampere University, Tampere, Finland

[*] These authors contributed equally to this work.
* teemu.ihalainen@tuni.fi

## Abstract

It is well established that mechanical cues, e.g., tensile- compressive- or shear forces, are important co-regulators of cell and tissue physiology. To understand the mechanistic effects these cues have on cells, technologies allowing precise mechanical manipulation of the studied cells are required. As the significance of cell density i.e., packing on cellular behavior is beginning to unravel, we sought to design an equiaxial cell compression device based on our previously published cell stretching system. We focused on improving the suitability for microscopy and the user-friendliness of the system. By introducing a hinge structure to the substrate stretch generating vacuum chamber, we managed to decrease the z-displacement of the cell culture substrate, thus reducing the focal plane drift. The vacuum battery, the mini-incubator, as well as the custom-made vacuum pressure controller make the experimental setup more flexible and portable. Furthermore, we improved the efficiency and repeatability of manufacture of the device by designing a mold that can be used to cast the body of the device. We also compared several different silicone membranes, and chose SIL-PURAN® due to its best microscopy imaging properties. Here, we show that the device can produce a maximum 8.5% radial pre-strain which leads to a 15% equiaxial areal compression as the pre-strain is released. When tested with epithelial cells, upon compression, we saw a decrease in cell cross-sectional area and an increase in cell layer height. Additionally, before compression the cells had two distinct cell populations with different cross-sectional areas that merged into a more uniform population due to compression. In addition to these morphological changes, we detected an alteration in the nucleo-cytoplasmic distribution of YAP1, suggesting that the cellular packing is enough to induce mechanical signaling in the epithelium.

## Introduction

Cells are constantly subjected to mechanical forces originating either internally, from adjacent cells or from the external environment. These forces include shear stress from fluid flow, tensile stress from increased cellular contractility, compressive stress from growth-induced

**Data Availability Statement:** The data relevant to this study are available from the Science Data Bank at the following DOIs: http://doi.org/10.57760/sciencedb.01741 http://doi.org/10.57760/

sciencedb.01742 http://doi.org/10.57760/
sciencedb.01743 http://doi.org/10.57760/
sciencedb.01744 http://doi.org/10.57760/
sciencedb.01764.

**Funding:** - P.K. - grant number 336785, the Academy of Finland, (https://www.aka.fi/en/) - The funders had no role in study design, data collection and analysis, decision to publish, or preparation of the manuscript. - T.O.I - grant numbers 308315 and 314106, the Academy of Finland (https://www.aka.fi/en/) - Emil Aaltonen foundation (https://emilaaltonen.fi/) - The funders had no role in study design, data collection and analysis, decision to publish, or preparation of the manuscript. - E.M. - grant number 332615, the Academy of Finland, (https://www.aka.fi/en/) - The funders had no role in study design, data collection and analysis, decision to publish, or preparation of the manuscript. - S.N. - grant number 326366, the Academy of Finland, (https://www.aka.fi/en/) - Emil Aaltonen foundation (https://emilaaltonen.fi/) - The funders had no role in study design, data collection and analysis, decision to publish, or preparation of the manuscript.

**Competing interests:** The authors have declared that no competing interests exist.

crowding, and finally, mechanical cues (e.g., stiffness, topography) arising from changes in the extracellular matrix [1–3]. Within cells, these forces and cues are sensed by a mechanotransduction process which converts the physical stimuli into biochemical and electrical signals [4–6]. Compressive forces have been shown to play important roles in cell and tissue homeostasis, development and cell differentiation [3, 7]. Intrinsic compressive forces co-regulate embryogenesis through affecting signaling pathways regulating gene expression [8] and via these mechanisms influence morphogenetic patterning [9]. In mature epithelium, a proper cell density is crucial for maintaining the barrier function of the tissue. The optimal cell density is maintained by sensing local stretch and compression and by adjusting proliferation and cell extrusion, respectively [1, 10]. Moreover, although the precise molecular pathways are unknown, it is commonly accepted that compression promotes stem cell differentiation into chondrocytes *in vitro* [11, 12]. Due to the last two decades of research, it is now acknowledged that compressive forces and flaws in the mechanotransduction pathways are significant co-regulators in pathological conditions such as cancer, affecting disease progression, heterogeneity, and metastasis [13, 14]. To gain better understanding of the associated biomechanical cellular mechanisms, platforms allowing robust mechanical stimulation of cells combined with different methodologies and readouts are required. Furthermore, as it is known that the direction of the applied force affects for example cellular orientation [15, 16], both equiaxial and uniaxial stimulation platforms are needed.

Platforms that allow mechanical stimulation of cells can be utilized to model diseases *in vitro* and to study mechanobiological signaling. As the use of mechanical stimulation has become more common, one of the main focus areas has been on cellular stretching, leading to the development of several stretching methods [17]. However, during the recent years the significance of compression has begun to unravel. Several devices exist that have been designed specifically for compression, but they mostly target vertical compression in 3D [18, 19] or 2D [13]. However, many stretching devices could, in principle, be utilized also for a compression experiment. The experiment modality, stretching or compression, defines the state of the substrate upon cell seeding. Culturing cells on a relaxed substrate will enable stretching, whereas culturing cells on a pre-strained substrate will allow the substrate strain to be released, thus compressing the cells. The most important requirement for compression is the possibility to maintain a static and stable stretch during the required cell culture period.

Most devices that fit these requirements have drawbacks in their suitability for live-cell imaging or for long term cell culture. The piezoelectric stretching device proposed by Deguchi and co-workers [20] and shape memory alloy -based device by Iwadate and Yumura [21] both produce uniaxial stretch and allow cells to be imaged with an inverted microscope through a thin membrane on which cells are growing. However, both devices contain electronic wiring which generates a safety hazard in humid cell culture conditions. Also, the piezoelectric device is bulky and therefore difficult to fit under a microscope, and the shape memory alloy is actuated with heat, which can cause problems in maintaining the correct temperature in long term cell culture. The pneumatic systems by Huh et al. [22], Huang and Ngyuen [23], and Pavesi et al. [24] all have a similar approach, where the cell culture chamber is lined with two vacuum chambers that uniaxially stretch the substrate on which cells grow. As these devices are based on pneumatic actuation, all electronics can be kept at a safe distance from the humid conditions required for cell culture. However, in all these devices high resolution microscopy is hindered because of multiple solid-liquid/gas interfaces located between cells and the objective being a significant drawback in many cell studies.

Devices for producing equiaxial strain also exist. Flexcell® inc. (Burlington, NC, USA) offers a variety of different stretching systems, even some that enable upright or inverted high resolution microscopy during strain (StageFlexer® and Inverted StageFlexer®). However,

these systems do not enable portability. The equiaxial stretching device reported by Quaglino [25] operates by a manual screw-top that pushes down an intender ring thus stretching the silicone membrane on which cells grow. As the device is free of any electronic parts, extended cell culture periods with static strain are possible. The small size of the device and unobstructed access to the cell substrate has allowed it to be used for confocal microscopy. However, as the device operates by pushing down the cell culture substrate, the z-directional movement of up to 6 mm makes loss of focus inevitable. The manual operation also rules out the possibility of cyclic stimulation or precise control over compression dynamics.

Previously, we have developed a robust mechanical manipulation device, Brick Strex, based on LEGO® bricks [26]. Brick Strex device can produce over 25% uniaxial strains for large cell populations and allows fluorescence microscopy in upright configuration. However, the device lacks direct incubation possibilities. Finally, IsoStretcher [27] contains a polydimethylsiloxane (PDMS) disc attached to six pins. A stepper-motor pulls the pins along a tangential trajectory thus stretching the central part of the PDMS disc equiaxially. The platform allows direct imaging of the thin PDMS disc on which cells grow and the stepper-motor enables automation and precise control over the strain dynamics. However, with the integrated stepper-motor the device is not suitable for long term cell culture.

Here, we have developed a PDMS-based pneumatic partial vacuum operated mechanical equiaxial stimulation platform. The device utilizes the same mode of action as the pneumatic stretching device [16, 28] but here we further develop the structure of the device and apply it to compression. The device enables an 8.5% equiaxial tensile strain to be stably and evenly applied to the entire cell culture substrate. As the partial vacuum pressure can be remotely controlled, the device can be kept inside a standard incubator in cell culture conditions for long periods of time. The device allows high resolution imaging with an inverted microscope as the device is accessible from below and even immersion objectives can reach through the 200 μm thick silicone membrane on which cells are cultured. The vacuum pressure controller enables cyclic strain, as well as static stretching or compression to be applied to the cell layer. The strain can be controlled with high repeatability and can be adjusted even during imaging due to remote control. Furthermore, our system enables portability, which makes it possible to transport the sample to the microscope without losing the pre-strain in compression applications. This simplifies arranging the experimental setup, as cell culturing and imaging can be done in separate facilities. Our device therefore stands out as one of the few mechanical stimulation platforms that enables both stretching and compression, that has excellent properties for high resolution imaging, allows portability and has high control over the dynamics of stimulation even during imaging. In this paper, we demonstrate the device to investigate the role of intercellular i.e., lateral compression in epithelial cell packing and physiology.

## Materials and methods

### Operation principle of the device

The device (Ø32 mm x 13 mm) consists of a cell culture well surrounded with a vacuum chamber, and of a 200 μm thick SILPURAN® (SILPURAN® FILM 2030, Wacker Chemie AG, Munich, Germany) silicone membrane that seals the vacuum chamber and acts as the cell culture substrate (Fig 1A). Partial vacuum pressure is applied to the vacuum chamber via a small inlet in the roof of the vacuum chamber and controlled with a custom-made vacuum operation system [28]. The pressure deforms the inner wall of the vacuum chamber leading to equiaxial pre-strain of the entire cell culture area (Fig 1B). Relaxing this pre-strain leads to lateral compression. The stabilizer ring (S1 Fig) supports the outer wall of the vacuum chamber and thus directs deformation to the inner wall.

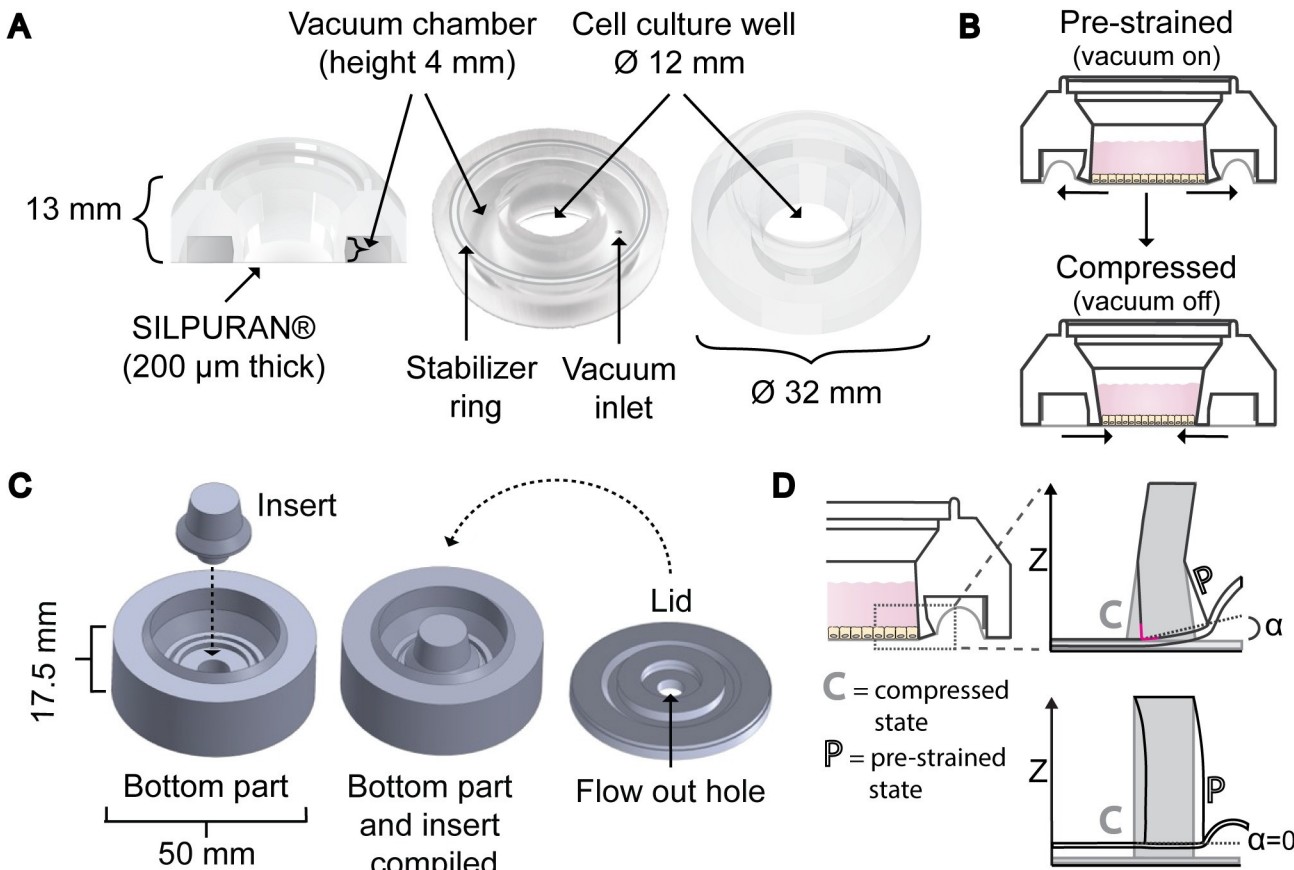

**Fig 1. Structure and function of the device and the mold.** (A) Structure and dimensions of the device illustrated as a cross section of the device (left), from below (center) and from above (right). (B) Diagram of the mode of action of the device. In the pre-strained state partial vacuum pressure in the vacuum chamber deforms the silicone membrane and pulls the inner walls of the cell culture well thus leading to equiaxial strain in the cell culture well. Once the pre-strain is released, the cell culture well is compressed. (C) The mold for the device consists of a bottom part, an insert, and a lid. The insert is detachable to allow an easy removal of the cured device. The hole in the lid allows excess PDMS to flow out. A weight was placed on top of the compiled mold during curing. (D) Schematic view of the inner wall of the vacuum chamber demonstrating how the hinge structure decreases z-directional movement of the silicone membrane. The hinge structure produces an angle α between the base of the inner wall and the horizontal plane. As the angle grows, the inner corner (marked in magenta) of the wall pushes the silicone membrane downwards thus compensating for the z-directional rising. Without the hinge structure (below), no angle is formed and therefore, the deformation of the wall causes the silicone membrane to rise. C refers to the compressed state of the wall, shown in grey, and P refers to the pre-strained state, shown with black boarders.

## Manufacturing of the device

The device is made of three parts: the body, the silicone membrane and the stabilizer ring. The body is made from PDMS and the SILPURAN® membrane out of silicone, both transparent and bioinert materials that can be sterilized by autoclaving. The stabilizer ring that is not in contact with cells or media was 3D-printed with Clear V4 resin (Formlabs, Somerville, MA, USA).

The body of the device is manufactured by mold casting PDMS. Compared to the previous punch-and-assemble approach [16, 28], the throughput, yield and repeatability in the device manufacture can be increased. The mold was designed with SOLIDWORKS® 2018 (Dassault Systèmes SolidWorks Corporation, Waltham, MA, USA) and 3D-printed (Keijom Oy, Pirkkala, Finland) with polyjet technology using Vero photopolymers (Object30, Stratasys Inc., USA) or machined from polyoxymethylene. It consists of a bottom mold, a removable insert, and a lid (Fig 1C). The bottom mold forms the outer face of the device, and the insert produces

the cell culture well. The insert also aids removal of the cured device from the mold as the device can be easily pushed out along with the insert. The lid forms the vacuum chamber of the device and the surfaces that are bonded to the silicone membrane. The hole in the middle of the lid enables excess PDMS and possible air bubbles to escape from the compiled mold during casting.

For casting the body of the device, PDMS was prepared as follows. The base elastomer and curing agent from SYLGARD™ 184 Elastomer Kit (Dow, Midland, MI, USA) were mixed in a ratio of 10:1. Vacuum was used to remove bubbles from the mixture before carefully pouring the PDMS into the assembled bottom mold and onto the lid with the hole sealed with tape. Vacuum treatment was repeated to PDMS-filled mold parts before rapidly placing the lid on the bottom mold. The tape was then removed to allow excess PDMS and air bubbles to escape. The filled molds were placed into an oven for curing (60°C, 10 h) and weights were placed on top to seal the molds tightly.

After the curing, the mold was disassembled to remove the body of the device. Residue PDMS was trimmed from the inner walls of the cell culture well with a punch and from the outer edges of the device with a scalpel. A G18 injection needle was sharpened into a punch and used to make a ~0.9 mm Ø hole to the ceiling of the vacuum chamber for the vacuum inlet. Prior to completing the device by bonding the silicone membrane to the body of the device, the body was washed with detergent to remove grease, rinsed with isopropanol and Milli-Q water and finally thoroughly dried. A Pico Low-pressure plasma system from Diener electronic GmbH + Co. KG (Ebhausen, Germany) was used to oxygen plasma bond (30 mbar, 30 W, 15 s) the device to the silicone membrane.

## Measuring strain

1 μm Ø Dragon Green fluorescent polystyrene microbeads (Bangs Laboratories, Inc., Fishers, IN, USA) were diluted 1:100 in $dH_2O$ and incubated for 1 h at room temperature (RT) in order to adsorb beads to the surface of the silicone membrane in the cell culture well. After incubation, excess beads were removed by gently rinsing with $dH_2O$. Devices were allowed to dry before imaging.

ZEISS Axio Scope microscope with A-Plan 10x/0.25 Ph 1 objective (ZEISS International, Oberkochen, Germany) was used to take image sequences of membrane deformation. The membrane was deformed by decreasing the pressure from -4 mbar to -350 mbar or to -600 mbar in 20 mbar or 75 mbar steps and a bead chosen for the analysis was centralized in all images. The microscope was manually refocused after each step. The z-coordinates were documented at each measurement point and used to determine the z-displacement. Based on careful visual inspection, we estimated a ±3 μm error arising from the manual refocusing. Within one strain measurement, three locations near the edge of the cell culture well and one location in the center were measured. Three to five repeated image sequences were taken at each location.

Prior to taking image sequences the device was leveled. This was done by manually finding the focus at eight points on the edges of the device. The microscope sample table was adjusted so that maximum 50 μm difference in leveling along the diameter occurred. This leads to maximum 0.239° angle compared to horizontal plane of the device, which, in turn, will lead to maximum 0.0009% error in x y-plane strain measurements. This error is so small that it can be neglected.

Particle Tracker 2D/3D plugin (Mosaic) [29] in Fiji [30] was used to track the trajectories of the beads. Five trajectories were chosen from opposing corners of the field of view to maximize the difference in x- and y-coordinates for minimizing the error. Beads were chosen carefully to avoid incorrect tracking, and trajectories at the edges of the field of view were preferred.

The relative change in distance between two trajectories was calculated with the following equation:

$$\left( \frac{\sqrt{\left(X_{ref,i} - X_{comp,i}\right)^2 + \left(Y_{ref,i} - Y_{comp,i}\right)^2}}{\sqrt{\left(X_{ref,-4mbar} - X_{comp,-4mbar}\right)^2 + \left(Y_{ref,-4mbar} - Y_{comp,-4mbar}\right)^2}} - 1 \right) * 100\%$$

where $(X, Y)_{ref,i}$ and $(X, Y)_{comp,i}$ mean the coordinates of the two particles compared in each calculation (named reference particle and comparison particle) at a measurement point i, and $(X,Y)_{ref(-4\ mbar)}$ and $(X,Y)_{comp(-4\ mbar)}$ mean the initial coordinates of the reference and comparison particles. Five trajectories in one corner of the image were compared to another five trajectories in the opposing corner thus leading to 25 calculations. The mean deformation and standard deviation (SD) were calculated for each measurement and plotted in Excel (Microsoft, Redmond, WA, USA). Mann-Whitney U test in SPSS® Statistics (IBM®, Armonk, NY, USA) was used to calculate the statistical difference between maximum strains across the four measurement spots.

## Optical resolution

Optical resolution of silicone membranes was determined by measuring the Full Width at Half Maximum (FWHM) of a point spread function (PSF) of an imaged fluorescent nanoparticle. We tested SILPURAN® and ELASTOSIL® (ELASTOSIL® FILM 2030, Wacker Chemie Ltd) in 100 μm and 200 μm thicknesses, Gloss (Gloss Sheeting, Specialty Manufacturing Inc., Saginaw, MI, USA) in 125 μm and 250 μm thicknesses and finally self-made 120 μm thick SYLGARD™ 184 membrane (S1 Appendix). The measurements were done with ZEISS LSM780 (ZEISS International) confocal microscope using 0.2 μm Ø fluorescent FluoSpheres™ (Invitrogen, Thermo Fisher Scientific, Waltham, MA, USA) beads. Beads were attached to the surface of the tested silicone membranes by simple physisorption. Beads were diluted 1:100 in $dH_2O$ and incubated for 1 h at RT. Before imaging, the excess beads were removed by gently rinsing with dH2O and devices were allowed to dry. Z-stacks were taken with C-Apochromat 40x/1.20 W Korr M27 (ZEISS International) objective in 200 nm intervals to cover the entire bead PSF. Fiji was used to create orthogonal projections of the resulting PSF and to plot the intensity profile (ten PSFs per sample). The data was normalized with the maximum intensity value of each bead, and the mean of these normalized z-direction intensity profiles was calculated. MATLAB (MathWorks®, Natick, MA, USA) was used to fit this profile into 1D-Gaussian distribution and to calculate the FWHM value. For orthogonal view images the z-stacks were resliced orthogonally and maximum intensity projections were taken from the slices covering the target beads. The images were cropped to contain only the target bead. Gaussian blur operation (3x3 kernel) was applied, and the brightness was adjusted so that equal intensity ranges were reached in the images.

## Cell culture

Cell culture wells of devices were coated with collagen I. Collagen I (rat tail, #A10483-010, Gibco, Thermo Fisher Scientific, Waltham, MA, USA) was diluted to 0.1 mg/ml in 0.02 N acetic acid and pipetted on devices so that the entire cell culture area was covered. Collagen was incubated for 45 minutes under UV-light and washed twice with Dulbecco's phosphate-buffered saline (DPBS). Approximately $2 \times 10^4$ Madin-Darby canine kidney type II (MDCK II) cells were seeded on the ~1.13 $cm^2$ cell culture area. In the experiments, either occludin-mEmerald or actin chromobody-mCherry expressing MDCK II cells were used. mEmerald-

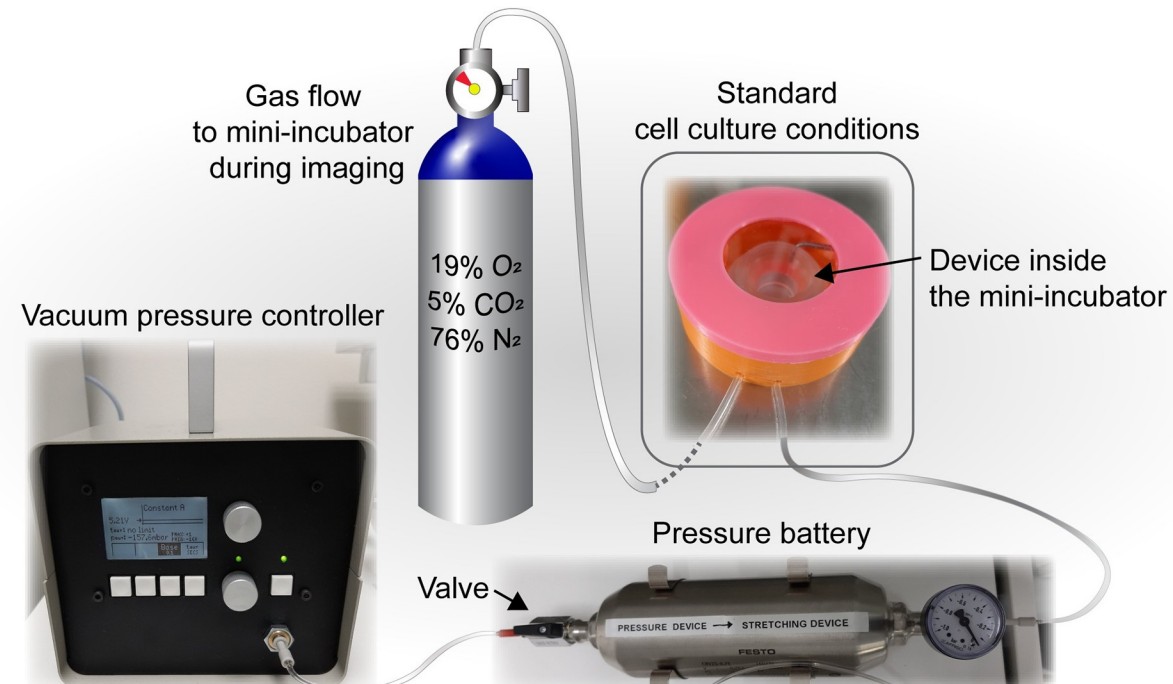

**Fig 2. The experimental setup.** The device was set inside the mini-incubator and connected to the vacuum pressure controller. The vacuum battery was attached between the device and the vacuum pressure controller. During the cell culture period the mini-incubator containing the device was kept inside a standard cell culture incubator. The valve of the vacuum battery was opened, thus allowing the vacuum pressure controller to maintain pressure during cell culture. After the six-day cell culture period during transportation to the microscope, the valve of the vacuum battery was closed, which allows the setup to be momentarily detached from the vacuum pressure controller as the vacuum battery maintains the vacuum pressure. At the microscope, the device was again connected to the vacuum pressure controller and the valve was opened. Gas flow (5% $CO_2$) to the mini-incubator was initiated once the device was removed from the standard incubator to maintain pH during transportation and imaging. Heated gas to maintain the temperature during the imaging was supplied into the microscope stage top incubator.

occludin-C-14 was a gift from Michael Davidson (Addgene plasmid # 54211; http://n2t.net/addgene:54211; RRID:Addgene_54211) and mCherry-actin chromobody was a kind gift from Research Director Maria Vartiainen (University of Helsinki, Finland).

Devices were placed inside the custom-made mini-incubator and connected to the vacuum pressure controller and vacuum battery (S1 Appendix). The experimental setup is illustrated in Fig 2. A radial pre-strain of 8.5% was initiated immediately after cell seeding. Cells were cultured in a standard cell culture incubator for six days until mature epithelium was formed. Minimum Essential Medium + GlutaMAX (#41090–028, Gibco) supplemented with 10% Fetal Bovine Serum (#10500–064, Gibco), 1% Pen Strep (#15140–122, Gibco) and 0.25 mg/ml G418 selection antibody (#04727894001, Roche, Basel, Switzerland) was used as culture media.

*Confocal imaging of compression.* Once the epithelium had matured for six days, samples were transported from cell culture to ZEISS LSM 780 (ZEISS International) confocal microscope. Gas flow to the mini-incubator was initiated once the device was removed from the standard incubator and continued throughout imaging to maintain correct pH in the media (Fig 2). The vacuum battery was used to maintain the pre-strain during transportation. Once at the microscope, the vacuum pressure controller was reconnected. Z-stacks were taken with C-Apochromat 63x/1.20 W Korr M27 (ZEISS International) objective before compression and after 15% areal compression in 30 min intervals for four hours. Several locations were imaged at each time point.

*Image analysis.* Tissue Cell Geometry Stats macro [31] in Fiji was used to segment cells and to determine cell cross-sectional areas from occludin-mEmerald expressing MDCK II cells.

MATLAB was used to generate histograms with 20 bins of each data set and histogram bin counts (mean cross-sectional area and number of cells per bin) were used to produce cross-sectional area distribution graphs. Additionally, the decrease in mean cell cross-sectional areas were calculated for each time point. Four experiments were analyzed.

For analysis of the cell height, z-stacks were resliced orthogonally, and summed intensity z-projections were taken. For images, the projections were taken only the slices covering a target cell, for measurements, entire stack was projected. Gaussian blur operation (5x5 kernel) was applied, and the intensity profiles were plotted with from apical to basal side. Line width was set to 100 μm to get an averaged profile plot. The widths of the intensity peaks were measured. Three experiments were analyzed.

*Immunostainings.* After the four-hour compression experiment was finished, cells were fixed with 4% paraformaldehyde for 10 min at RT, following two washes with DPBS. Cells were permeabilized with 0.5% Bovine Serum Albumin (BSA) and 0.5% Triton-X in DPBS for 10 minutes and blocked with 3% BSA DPBS for 1 h. YAP1 antibody (clone 63.7, sc-101199, Santa Cruz Biotechnology, Dallas, TX, USA) was diluted 1:300 in 3% BSA-DPBS and incubated at RT for 1 h. 10-minute washes with permeabilization buffer, DPBS and permeabilization buffer were performed before adding the secondary antibody for 1 h (goat anti-mouse, A488, #A11029, Invitrogen, 1:200 in 3% BSA-DPBS). Samples were washed twice with DPBS for 10 minutes and with dH$_2$O for 5 minutes. The cell culture area was detached from the device with a scalpel and placed cells upwards on an objective glass. ProLong™ Diamond Antifade Mountant with DAPI (Invitrogen) was pipetted on top of the cell layer and a coverslip was mounted on top. Control devices were prepared similarly. Samples were protected from light during all incubations to prevent bleaching.

*Confocal imaging and image processing of YAP1-stained samples.* Mounted samples were imaged with laser scanning confocal microscope Nikon A1R mounted in inverted Nikon Ti-E body (Nikon, Tokyo, Japan). Plan Apo VC 20x DIC (Nikon) objective was used to image a large field of view. Z-stacks were taken at several locations of each sample using 405 nm, 488 nm, and 561 nm channels for DAPI, occludin-mEmerald and YAP1, respectively. Several representative areas containing about 25 cells from each image were analyzed. The analysis was performed to the z-slice crossing at the center of the nuclei. By using the DAPI-channel, the nuclei were segmented with Otsu thresholding. In addition, 1 μm thick bands were expanded around the segmented nuclei. These regions of interest were used in the YAP1-channel to measure the mean signal intensity from the nucleus and from the cytoplasm (band surrounding the nucleus) of each cell. The ratio between the nuclear and cytoplasmic value were determined for each analyzed cell. Independent samples Mann-Whitney U-test SPSS® Statistics was used to calculate the statistical difference between compressed and control samples.

*Numerical modelling and simulation of the device.* Finite element method (FEM) was implemented using COMSOL Multiphysics 5.4 (COMSOL, Inc., Burlington, MA, USA). The device was simulated in a two-dimensional (2D) symmetric model (S1 Fig) using Solid Mechanics module. PDMS and SILPURAN® were modeled as hyperelastic materials using Neo-Hookean model with the following parameters. For PDMS, density 971 kg/m$^3$, Poisson's ratio 0.499, Young's modulus 2x10$^6$ Pa and Lamé parameter 6.67x10$^5$ N/m$^2$, and for SILPURAN®, 1080 kg/m$^3$, 0.499, 1.05x10$^6$ Pa and 6.67x10$^5$ N/m$^2$, respectively. Values from literature were used for PDMS, whereas SILPURAN® parameters were obtained from the manufacturer. It was assumed that the 3D printed stabilizer ring underwent negligible deformation and therefore, it was modeled as Rigid Domain in the simulations. The model consisted of approximately 17000 elements and was solved using stationary direct solver. Simulations with different pressures between 0 to -520 mbar were solved using auxiliary sweep option with a step size of 500 Pa.

## Results

### Functional design and characterization of the device

We started the device construction by first simulating the device performance. Based on the COMSOL simulations, we did several upgrades to the original device [16, 28]. The structure was redesigned so that the body of the device could be cast from a single 3D mold (Fig 1C), allowing a more streamlined manufacturing process. In addition, we made modifications to reduce the movement along the imaging axis (z-directional movement) of the silicone membrane to further improve imaging properties. To achieve reduced z-directional movement of the silicone membrane, we modified the inner wall of the vacuum chamber by introducing a hinge structure (Fig 1D). The mode of action of the device intrinsically creates z-directional movement to the silicone membrane i.e., the cell culture substrate, thus leading to loss of focus during microscopy imaging. The hinge structure decreases this movement by targeting the deformation of the wall to the hinge, which in turn forces the base of the inner wall to tilt. As the base of the wall tilts, the inner corner of the wall pushes the silicone membrane downwards, and therefore compensates for the lifting of the membrane [32]. The slightly thicker base of the wall further increases the compensation effect.

As said, simulations were used to upgrade the structure of the device and the numerical model was experimentally validated by measuring the deformation of the membrane using different supply of partial vacuum pressures. When experimental and simulated strain data were compared, a close resemblance up to ~-350 mbar was observed as shown in Fig 3A and 3B. Within this range, strain is relatively linear up to ~-200 mbar (linear fit $R^2$ = 0.98) but continues to increase up to -350 mbar. Strain was measured both as negative pressure was increased from -4 mbar to -350 mbar (line called stretch in Fig 3A and 3B), and as pressure was returned to the initial -4 mbar (line called compression in Fig 3A and 3B). These lines concur, demonstrating that hysteresis does not occur within this strain range. Furthermore, the strains are similar in the center of the device (Fig 3A) and in the edges of the device (Fig 3B) suggesting that the device creates a uniform strain field.

The performance of a single device was measured three times to verify the repeatability of its behavior. Between the first and second measurement the device was in 1 Hz cyclic strain for 6 days and between the last two measurements in rest for 5 days. The mean radial strain at -350 mbar from four measurement points was 8.2 ± 0.3% (mean ± SD), suggesting that neither stress nor shelf life affect device performance (Fig 3C). Additionally, we wanted to analyze the device-to-device variation arising from the manufacturing process. To this end, several devices (n = 6) were analyzed from four measurement points. The mean maximum strain (-350 mbar) measured from six devices was 8.5 ± 0.6% indicating a low variation arising from the manufacture (Fig 3D). The strain is significantly similar across the four measurement points (p-value = 0.978, ANOVA), thus indicating even strain throughout the cell culture area. As the maximum radial pre-strain of 8.5% is released, a relative 7.8% radial compression occurs ($\frac{1-1.085}{1.085} = -0.078$). This corresponds to a 15% areal compression of the cell culture well. We also determined the effect of cell culture media to device performance (S1 Fig) and saw that media does not affect the maximum strain.

With the hinge structure, we managed to reduce the total z-displacement by 86% to 45 ± 22 µm from the initial 315 ± 22.5 µm [28]. The silicone membrane first descends as the pressure decreases to -150 mbar, after which it starts to return to the zero-stage (Fig 3E and 3F). This is caused by the hinge structure, which slightly overcompensates for the rising of the membrane at smaller strains. In repeated measurements on a single device the z-displacement was systematic, indicating that the measurement error is small, and that devices function with high repeatability (Fig 3F). However, when the z-displacements of six different devices were

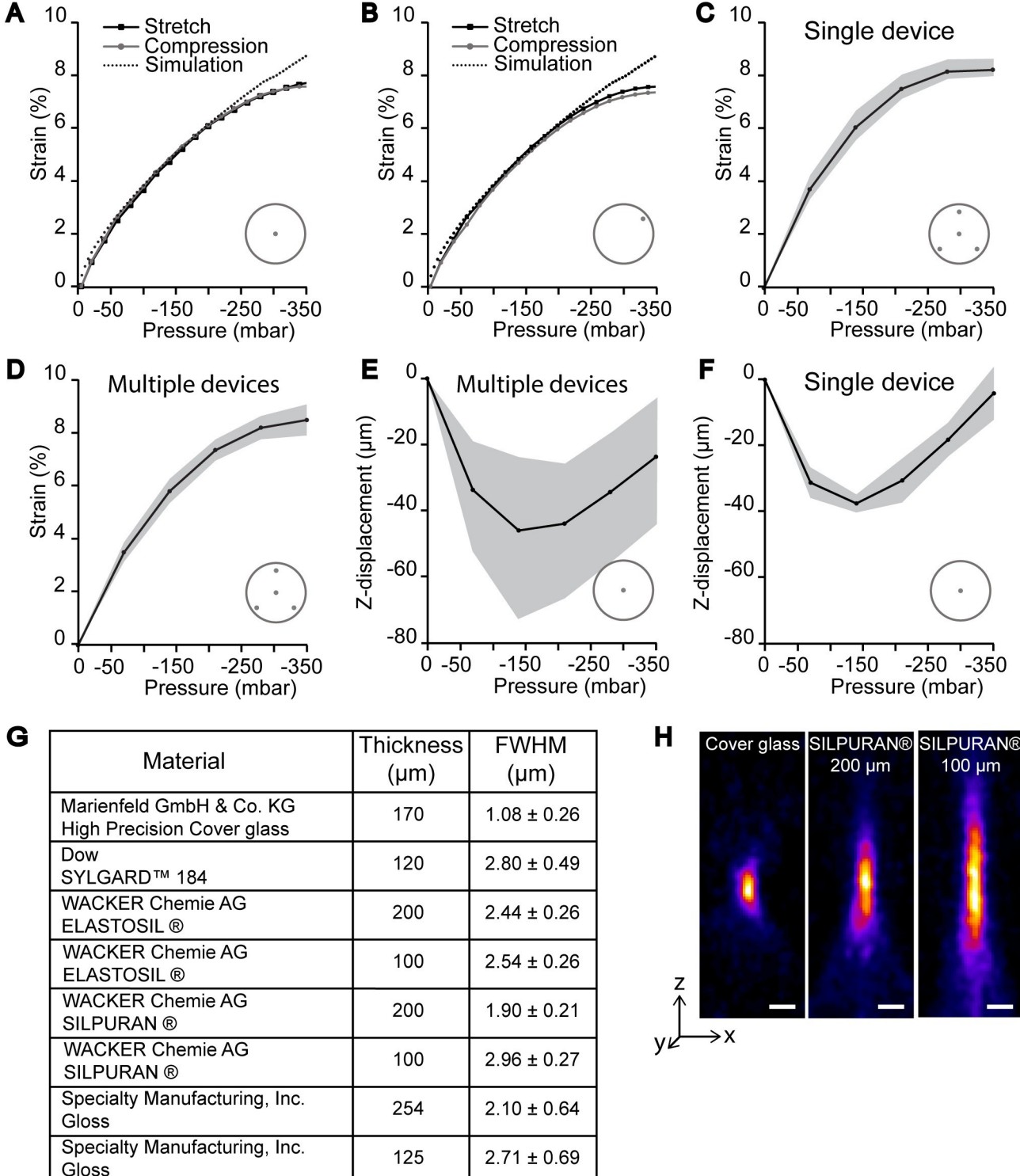

**Fig 3. Functional characterization of the device.** A) Pressure-strain graph of simulation data (dashed line) compared to experimental data measured from the center of the device and (B) edge of the device. Strain was measured as pressure was decreased (stretch) and as it was released (compression) to detect possible hysteresis (C) Pressure-strain graph showing mean and SD (grey area) of repeated measurements (n = 3) performed on a single device from four measurement locations. (D) Pressure-strain graph showing mean and SD (grey area) of measurements performed on six different devices. Each device was measured from four measurement locations. (E) Mean and SD (grey area) of the movement of the device membrane along the imaging axis (z-direction) in the center of the device (n = 6). (F) Mean and SD (grey area) of the z-directional movement in the center of a single device measured repeatedly (n = 3). G) Table with FWHM values along the z-axis for cover glass and different silicone membranes. 200 μm thick SILPURAN® has the

lowest FWHM value out of the compared silicone membranes indicating best imaging properties. The manufacture of 120 μm thick SYLGARD™ membrane is supplied in S1 Appendix. (H) Orthogonal (xz-plane) projections of 0.2 μm Ø fluorescent FluoSpheres™ imaged through High Precision cover glass (170 μm) and SILPURAN® silicone membranes of 200 μm and 100 μm thicknesses. The material affects especially the z-directional resolution. Scale bar 1 μm. In (A)-(F) the dot(s) inside the grey circle illustrate the measurement location within the cell culture well.

compared, the SD increased notably. This indicates that the z-displacement is sensitive to small deviations in the device geometry that arise from the manufacturing process.

The device geometry creates no obstacles between the cell culture substrate and the objective, thus enabling imaging with an inverted microscope during compression. Furthermore, the membrane thickness of 200 μm allows the use of high magnification and numerical aperture objectives. Owing to the vacuum battery, the strain can be maintained temporarily even without the vacuum pressure controller. The mean pressure leak during 72 hours was measured to be 6.1 mbar/h (S1 Appendix). To minimize any pressure leak during transfer to the imaging system, only a short transfer time of 30 min was used preserving 99% of the pressure and ensuring strain stability. Additionally, a custom-made mini-incubator ensures that a stable 5% $CO_2$ atmosphere can be maintained even during extended imaging periods and during transportation (S1 Appendix). After 24 h with 7 ml/min gas flow the pH of cell culture media remained 7.275.

Next, we tested different silicone membranes at the confocal microscope to optimize the high-resolution imaging properties of the device. Optical performances of the membranes were quantified by imaging fluorescent nanoparticles adhered to the membrane surface. The confocal 3D-image stacks of the nanoparticles can be used as an approximation of the point spread function (PSF) of the system. The performances of the different membranes were compared to the optimal imaging substrate, 170 μm thick high-precision coverslip glass. All membranes caused strongly elongated PSFs when compared to the glass coverslip (Fig 3F and 3G). The FWHM of the PSF in the z-direction were determined for three commercial silicone membranes and one self-made PDMS membrane (S1 Appendix) in varying thicknesses. Out of the candidates, SILPURAN® 200 μm had the smallest FWHM value which means that it causes the smallest z-directional aberration, and therefore results in the best optical resolution. However, no difference was seen in the in-plane (xy-directional) resolutions, suggesting that the membrane causes axial spherical aberration. We also confirmed that the autofluorescence from SILPURAN® was insignificant for imaging purposes (S2 Fig and S1 Appendix). Thus, due to its good optical properties, SILPURAN® 200 μm was chosen for the manipulation experiments.

## Effects of lateral compression on epithelium morphology

MDCK II cells stably expressing occludin-mEmerald were cultured in collagen I coated devices. Pre-strain (-350 mbar) was initiated before the cells adhered to the substrate. After six days in culture in pre-strained devices, the cells formed a dense epithelial monolayer with mature tight junctions and hexagonal cell shape as detected in Fig 4A. In live-cell compression experiments, the cells in the device were imaged by taking optical z-stacks before compression as a reference followed by relaxation of the pre-strained membrane. The cells were then imaged immediately after the compression followed by imaging in 30-minute intervals up to four hours post-compression. Cross-sectional areas of the cells were determined before compression and at time points post-compression by detecting the cell vertexes from the occludin-mEmerald signal. In four out of five repetitions of the experiment, we saw a change in the cell cross-sectional area distribution before and after compression (Fig 4B). Before compression, two distinguishable peaks with cross-sectional area maxima at 82.4 μm$^2$ and 135.4 μm$^2$ were

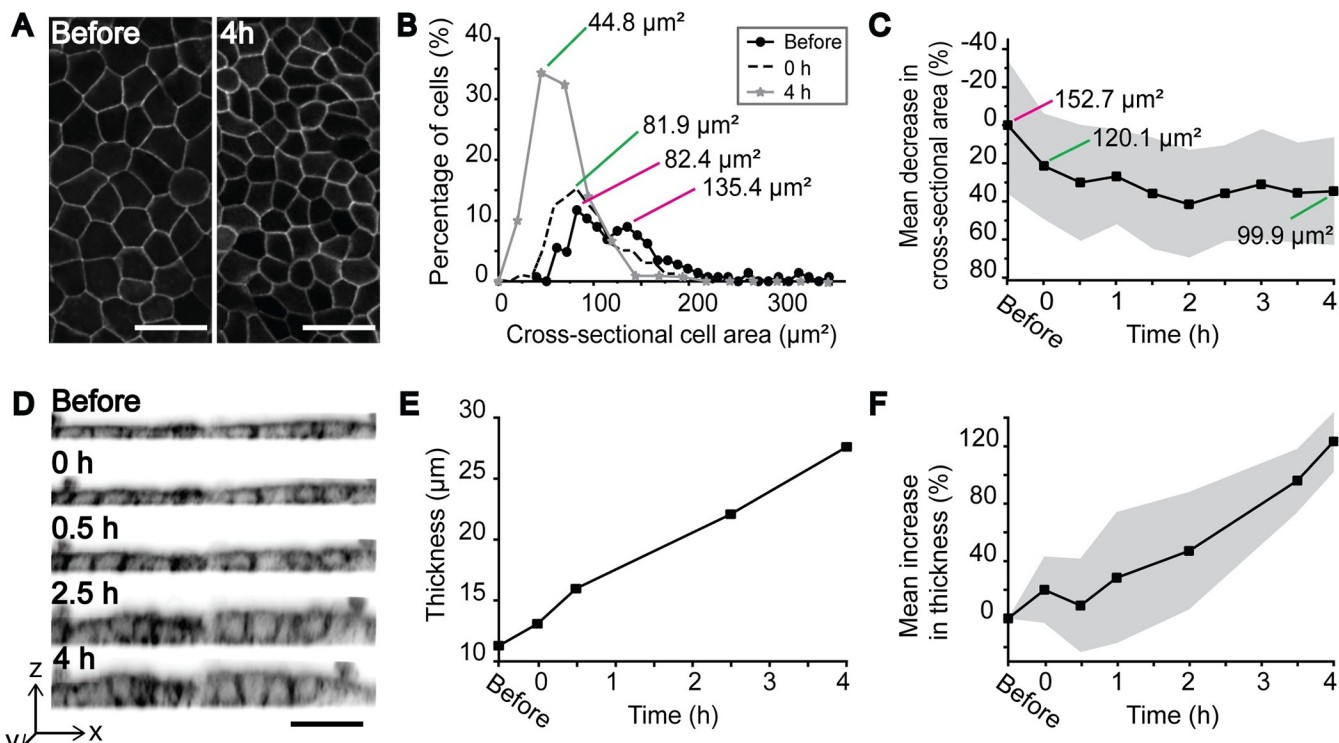

**Fig 4. Morphological changes caused by the lateral compression.** (A) Maximum intensity projections of occludin-mEmerald expressing MDCK II cells growing on the device, imaged before and 4 h after compression. Scale bar 30 µm. (B) Example case of the distribution of cell cross-sectional areas before compression, directly after compression and 4 h after compression. Before compression two populations with differing cell cross-sectional areas (marked with magenta lines) are clearly distinguishable. After compression, the distributions are more uniform, indicating of a more homogenous population, and cross-sectional area is smaller (green lines). (C) Mean and SD (grey area) of the decrease in cross-sectional area as a function of time (n = 4). (D) Orthogonal projections of the actin chromobody-mCherry expressing MCDK II cell layer at different time points showing packing and thickening of the cells (scale bar 30 µm, inverted gray scale LUT), and (E) corresponding cell layer thicknesses during compression. (F) Relative increase in cell layer thickness together with its SD (grey area) as a function of time (n = 3).

detected. This suggests that two populations with differing cross-sectional areas existed within the epithelial sheet. In contrast, immediately after the compression, these two separate populations became less evident, detected as a more homogenous cross-sectional distribution histogram. At the four-hour timepoint the difference in the cell cross-sectional area distribution is even more striking.

Concomitantly, compression produced a population wide decrease in cell cross-sectional areas (Fig 4A). Fig 4B shows how the peak of the histogram shifts toward smaller cross-sectional areas due to compression. Furthermore, as shown in Fig 4C, the mean cell cross-sectional area gradually decreases from 152.7 µm$^2$ to 99.9 µm$^2$ during the four hours following the compression. Directly after compression, when the response is purely mechanical, the mean decrease in cross-sectional area was 21%.

Cells are 3D objects and the reduction in the cross-sectional area is possible only if the cells shrink or if their height is increased. Cell shrinking would require rapid outflux of water from the cells. We assumed that the cell volume could not change that rapidly, and therefore suspected that as cross-sectional area decreases, the cell height must increase. Since occludin localizes mainly to tight junctions at the apical cell-cell interface, we decided to analyze the cellular height by using MDCK II cells stably expressing actin chromobody-mCherry. The compression was performed as described above and the height of the cell layer was analyzed by making orthogonal sections from the optical z-section series (Fig 4D and 4E). Indeed, we saw

a ~20% increase in the mean cell height directly after the compression, which concurs with the decrease in cell cross-sectional area. Likewise, an additional slow increase in thickness is seen during the four-hour experiment. (Fig 4F).

## Relocalization and cytoplasmic retention of YAP1 in response to increased lateral compression

Our compression experiments showed that the morphology of the epithelium is affected by the increased lateral compression. In epithelium, the morphological changes in respect to cell density are known to affect the activity of the Hippo-pathway [33]. In epithelium, the Hippo-pathway mediates contact inhibition of growth by acting via mechanosensory proteins including transcription regulator Yes-associated protein (YAP1) [34]. The intracellular localization of YAP1 is known to reflect the current activity of growth signaling. Low cell density, high substrate stiffness and cytoskeletal tension have been shown to lead to nuclear localization of YAP1 and thereby activation of proliferation inducing genes. In contrast, high cell density, low substrate stiffness and low cytoskeletal tension lead to nuclear exclusion and deactivation of YAP1 [35]. We have previously shown that 2 h after 20% uniaxial lateral compressive strain of MDCK II cells, YAP1 is excluded from the nucleus [26]. This suggests that the YAP1 related signaling is attenuated after epithelial compression.

Here, we immunostained both the static controls and compressed samples for YAP1 (Fig 5A). As Fig 5B shows, the nucleo-cytoplasmic fluorescence intensity ratio in the un-compressed sample was 0.99 ± 0.08 (mean ± SD). In contrast, after the compression YAP1 was localized into the cytosol with the mean nucleo-cytoplasmic ratio of 0.83 ± 0.06. According to independent samples Mann-Whitney U-test, there was significant statistical difference between the control and the compressed sample (p<0.0001). This experiment shows that as a response to lateral compression, YAP1 is excluded from the nucleus suggesting for cellular packing-induced inhibition of proliferation and growth-related signaling.

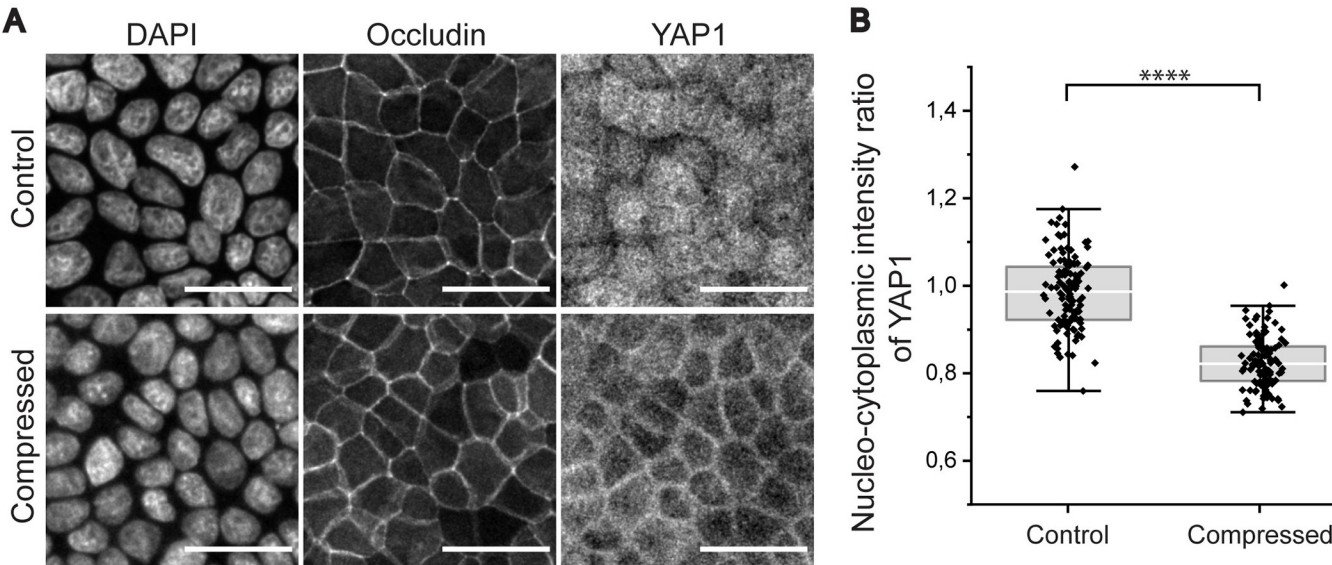

**Fig 5. Nucleo-cytoplasmic distribution of YAP1.** (A) Immunofluorescence staining of YAP1 together with DAPI and occludin-mEmerald. Static control above and compressed sample below. Scale bars 30 μm. (B) Nucleo-cytoplasmic ratio of YAP1 signal in static controls and compressed samples. P-value < 0.0001 (independent samples Mann-Whitney U-test).

## Discussion

It is well established that cellular physiology is co-regulated by physical cues, forces, and cellular mechanics [3, 5]. Cells can sense different forces (e.g., tensile, compressive, shear) in a mechanotransduction process, where physical force is converted into biochemical activity of the cells [2]. To understand these processes at the molecular level, there is a great need for tools and devices which allow precise mechanical manipulation of cells and microtissues. In this regard, tensional or compressive strain represents an interesting and physiologically relevant mechanical exercise. Especially epithelial cells can experience large changes in the strain, e.g., in our lungs during breathing and in our digestive system [36, 37]. Furthermore, during cancer development the cellular density is usually increased in the primary tumor [13, 38]. In this paper, we wanted to investigate the effect of cellular packing in epithelium. Therefore, we developed a pneumatic device, which can be used to induce 2D equiaxial mechanical compression to cell cultures. Similar design has been previously used to study cardiomyocyte differentiation by applying equiaxial [28] and uniaxial [16] cyclic stretch. The device proposed here can be used to induce compressive stress by culturing cells on a pre-strained substrate, thus leading to compression as pre-strain is released.

In many custom-made devices, manufacturing remains a bottle neck and difficulties to construct enough devices can hamper and delay research. Therefore, another aim of this study was to redesign the device such that it can be cast from a single 3D mold while maintaining the proven mode of action. The updated manufacturing process of the device is reproducible, and simple enough to be performed without special equipment. However, this 3D mold-based design also enables multi-cavity injection molding and therefore large-scale production. Furthermore, we envision that the system can be easily expanded to multiplate format simply by branching the pressure tubing. We also focused on minimizing the z-directional movement of the silicone membrane to further improve imaging properties. This was achieved by applying a hinge-structure in the inner wall of the vacuum chamber [32]. In conclusion, while re-purposing the device into enabling equiaxial lateral compression, we also improved its stability in microscopy by minimizing movement of the stretched membrane along the imaging axis and streamlined the manufacturing process.

The device has several advantages that make it a potential candidate in cell mechanobiology studies. Imaging can be performed directly on the device during stimulation as the cell culture substrate is easily accessible from below. The membrane thickness (200 μm) enables the use of high-resolution microscopy, and the remote actuation enables strain to be adjusted without disturbing the device and the cells in the device. Due to the remote actuation, devices can be kept safely in a standard incubator even during extended cell culture periods. The portability enabled by the vacuum battery and the mini-incubator permits the device to be transported between different equipment and laboratories without losing the membrane pre-strain. We also upgraded the experimental setup by building a designated vacuum pressure controller to replace the computer-controlled system. This further facilitates the use and portability of the device. Thus, the device enables high resolution live-cell imaging with an inverted microscope, is suitable for long term cell culture, and provides portability.

Based on the experiments, the proposed device can be used to produce an 8.5% radial pre-strain the release of which leads to a 15% equiaxial compression of the cell culture area. Variation among devices was low (measured radial strains 8.5 ± 0.6%) showing that the manufacturing process is repeatable. Furthermore, repeated testing on a single device showed that the device performance is not altered in use (measured radial strains 8.2 ± 0.3%). The strain is uniform throughout the entire cell culture substrate and no hysteresis occurs. The performance of our device is therefore robust and repeatable.

The 8.5% maximum strain enabled by our device is within a biologically relevant range. Strains ranging from 4% to 10% have been reported to affect cell proliferation, protein expression and inflammatory mediator expression in tendon fibroblasts and periodontal ligament fibroblasts [3]. Up to 10% strains have also been shown to promote mesenchymal stem cell differentiation [39] and cardiomyocyte orientation [16]. In addition to the degree of compression, also the speed and duration of deformation affect cell response. For example, applying 5% cyclic uniaxial stretching to tendon fibroblasts for one day increased proliferation, whereas applying the same stimulation for two days decreased proliferation [3].

As seen in Fig 3C and 3D, there are some minor discrepancies between experimental data and simulated data. This is caused by the hyperelasticity of PDMS [40] which creates challenges in simulating the mechanics of the device especially in higher strains. In experimental data the strain begins to saturate as pressure decreases below -250 mbar, which results in a narrower dynamic range, and subsequently in a lower maximum strain. Our hypothesis for this saturation is a contact that emerges between the silicone membrane and the inner walls of the vacuum chamber when the silicone membrane is deformed. We were able to visually observe a contact forming especially between the outer wall of the vacuum chamber and the silicone membrane. However, we were not fully able to produce the same effect in simulations. We did, however, observe that when we force a contact to be formed between the silicone membrane and the ceiling of the vacuum chamber in simulations, a saturation of stretch does indeed occur (S1 Fig). Albeit this is an artificial simulation condition, the result supports our hypothesis. We therefore suggest that widening the vacuum chamber would remove this contact formation and prevent strain saturation. Another feasible solution would be to apply a double layer of silicone membrane beneath the vacuum chamber. A thicker membrane lining the vacuum chamber could prevent contact formation, while the stretching and imaging properties of the silicone membrane lining the cell culture well would remain unchanged.

Finally, we show that the 15% areal compression is enough to spark a mechanobiological response in MDCK II epithelial cells. Compression alters the cellular morphology in three ways; it leads into an increase in homogeneity in cross-sectional areas, a decrease in mean cross-sectional area and an increase in cell height. Before compression, two predominant cell populations of differing cross-sectional areas were seen as two peaks or as a peak and a shoulder in the distribution curve of the cell cross-sectional areas (Fig 4B). After compression, these populations merged into one more homogenous population. This strongly suggests that the degree of cell shape remodeling is not homogenous throughout the cell population, and that a population of cells exists that undergo greater remodeling than the rest of the cells. One possible phenomenon explaining this finding is the super-elastic behavior of epithelial sheets. Latorre et al. [41] showed that constant tension under stretching is sustained by individual super-stretching cells instead of a homogenous cellular response. The compression-induced disappearance of the small population of larger cells could be a sign of this same event in reverse. As super-elasticity is caused by dilution of the actin cortex, such cells might be more susceptible also to compression. Therefore, the individual cells that would undergo extreme stretching in high strain could also be responsible for bearing compressional force by decreasing their cross-sectional area at a greater extent than other cells.

In addition to becoming more homogenous, the mean cross-sectional area of the cell population decreased. This is seen in Fig 4C, where the mean cell cross-sectional area decreases 21% directly after compression, and finally reaches a 35% decrease after four hours. Interestingly, cell cross-sectional areas seem to decrease more than the 15% compression supplied by the device. We suspect that this phenomenon is partly caused by the fact that actomyosin contraction as well as occludin, which was used for determining the cell cross-sectional areas, are apically located [42]. Therefore, cell cross-sectional areas were determined from the part of the

cell that is highly contractile. This apical contraction would lead to cells gaining wedge-like shapes, which in turn would cause the epithelium to curve. However, this possible curvature was not detected as imaging was done in such a small part of the entire epithelium. Furthermore, in the slower responses, seen during the 4 hours after compression, the direct mechanical effects may be accompanied by cellular adaptation [43], thus explaining the unexpectedly high change in mean cell area.

As epithelial cells maintain a constant volume during deformation [44, 45], we respectively saw an increase in the mean height of the cell layer as expected (Fig 4F). Compression therefore caused cells to adapt a more columnar or wedge-like shape. The purely mechanical response directly after compression produced a 20% increase in thickness, which concurs with the decrease in cell cross-sectional area. Again, the slower changes in cell heights can be accounted for by biological responses, for example osmotic swelling [43]. Furthermore, the large variation in cell responses also might affect both the decrease in mean cell cross sectional areas and increase in mean cell layer thickness.

Finally, we wanted to investigate whether the morphological changes of the epithelium were followed by cellular mechanosignaling. Therefore, we immunostained cells against YAP1, a well-established marker of mechanobiological activity. Cells were seeded on the prestrained devices and cellular YAP1 localization was investigated after compression. As seen in Fig 5A, YAP1 localized more into the nuclei in control samples, which indicates to active YAP1-signaling and therefore active proliferation. However, in compressed samples YAP1 was more cytoplasmic. In these samples, the cell density is increased and thus one can speculate that cell proliferation is not needed to maintain the epithelium homeostasis. These findings are in line with previous reports showing the effects of increased cell density i.e., cellular packing or lateral compression on intracellular localization of YAP1 [26, 34].

In this paper, we introduced a portable pneumatic equiaxial cell compression device reaching 15% areal compressive strains. The device can be mounted on an inverted fluorescence microscope and together with the SILPURAN® silicone membrane allows even high resolution live-cell imaging with immersion objectives. Thus, the device is a potential tool for mechanobiological research. We demonstrated the usability of the system for mechanical manipulation of cells by applying lateral compression and thus inducing increased packing of epithelial monolayers. The packing leads into increase in homogeneity in cross-sectional areas of the cells, reduction in the mean cell cross-sectional area, increase in the cellular height, and finally, relocalization and cytoplasmic retention of YAP1. To conclude, the presented device serves as a potential platform for studies on cell mechanics enlightening the effects of lateral compression for epithelial cell packing and physiology.

## Supporting information

**S1 Fig. Effect of contact formation to device function.** (A) Normal geometry used for simulations and a heat map of strain at -350 mbar. The stabilizer ring is marked with an arrow. (B) Artificial geometry where a pilar (marked with an arrow) is inserted below the vacuum chamber ceiling to force contact formation between the silicone membrane and the inner ceiling in the vacuum chamber. Heat map of strain at -350 mbar. (C) Simulated stress-strain data with the geometries presented in A and B. Forced contact causes strain to begin to saturate at -300 mbar. D) Comparison of strain measurements performed on dry devices (data from Fig 3D) and devices containing liquid. Strain data from liquid filled devices falls under the standard deviation region of devices measured dry.
(TIF)

**S2 Fig. Autofluorescence.** The excitation-emission-intensity plots for (A) Quartz, (B) High Precision cover glass, (C) Flexcell® membrane, (D-F) Polystyrene cell culture plastic, (G) SIL-PURAN®, (H) ELASTOSIL®, (I) Gloss, and (J) self-made SYLGARD™ membrane. (K) Table with maximum intensity values for each sample and the corresponding excitation and emission wavelengths. Note that Flexcell® membrane has a different intensity axis.
(TIF)

**S1 Appendix. Manufacture of SYLGARD™ membrane, vacuum battery characterization, mini-incubator characterization, and autofluorescence measurements.**
(DOCX)

# Acknowledgments

The authors acknowledge the Biocenter Finland (BF) and Tampere Imaging Facility (TIF) for the service. The authors acknowledge Elina Vuorimaa-Laukkanen and Nikita Durandin (Faculty of Engineering and Natural Sciences, Tampere University) for assistance with autofluorescence measurements and result interpretation. Professor Michael Davidson and Research Director Maria Vartiainen are gratefully acknowledged for mEmerald-occludin and mCherry-actin chromobody plasmids, respectively.

# Author Contributions

**Conceptualization:** Joose Kreutzer, Soile Nymark, Pasi Kallio, Teemu O. Ihalainen.

**Data curation:** Joose Kreutzer.

**Formal analysis:** Joose Kreutzer, Antti-Juhana Mäki.

**Funding acquisition:** Pasi Kallio, Teemu O. Ihalainen.

**Investigation:** Heidi Peussa, Joose Kreutzer.

**Methodology:** Joose Kreutzer, Antti-Juhana Mäki, Teemu O. Ihalainen.

**Project administration:** Pasi Kallio, Teemu O. Ihalainen.

**Supervision:** Joose Kreutzer, Elina Mäntylä, Soile Nymark, Pasi Kallio, Teemu O. Ihalainen.

**Validation:** Joose Kreutzer.

**Visualization:** Heidi Peussa, Antti-Juhana Mäki.

**Writing – original draft:** Heidi Peussa, Antti-Juhana Mäki.

**Writing – review & editing:** Heidi Peussa, Joose Kreutzer, Elina Mäntylä, Antti-Juhana Mäki, Soile Nymark, Pasi Kallio, Teemu O. Ihalainen.

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
