## [Decision Letter · Decision Letter 0]

31 Mar 2022

PONE-D-22-04502Pneumatic equiaxial compression device for mechanical manipulation of epithelial cell packing and physiologyPLOS ONE

Dear Dr. Ihalainen,

Thank you for submitting your manuscript to PLOS ONE. After careful consideration, we feel that it has merit but does not fully meet PLOS ONE’s publication criteria as it currently stands. Therefore, we invite you to submit a revised version of the manuscript that addresses the points raised during the review process.

Please, address all the comments made by reviewers, especially those related to the utility and novelty of this experimental setup. Please submit your revised manuscript by May 15 2022 11:59PM. If you will need more time than this to complete your revisions, please reply to this message or contact the journal office at plosone@plos.org. Please include the following items when submitting your revised manuscript:A rebuttal letter that responds to each point raised by the academic editor and reviewer(s). You should upload this letter as a separate file labeled 'Response to Reviewers'.A marked-up copy of your manuscript that highlights changes made to the original version. You should upload this as a separate file labeled 'Revised Manuscript with Track Changes'.An unmarked version of your revised paper without tracked changes. You should upload this as a separate file labeled 'Manuscript'.

We look forward to receiving your revised manuscript.

Kind regards,

Antonio Riveiro Rodríguez, PhD

Academic Editor

PLOS ONE

Journal Requirements:

Reviewers' comments:

Reviewer's Responses to Questions

**Comments to the Author**

1. Is the manuscript technically sound, and do the data support the conclusions?

Reviewer #1: Yes

Reviewer #2: Partly

Reviewer #3: Yes

2. Has the statistical analysis been performed appropriately and rigorously? 

Reviewer #1: Yes

Reviewer #2: No

Reviewer #3: Yes

3. Have the authors made all data underlying the findings in their manuscript fully available?

Reviewer #1: Yes

Reviewer #2: Yes

Reviewer #3: Yes

4. Is the manuscript presented in an intelligible fashion and written in standard English?

Reviewer #1: Yes

Reviewer #2: Yes

Reviewer #3: Yes

5. Review Comments to the Author

Reviewer #1: In this paper, the authors designed an equiaxial cell compression device which is suitable for microscopy. The device can produce a maximum 8.5% radial pre-strain which leads to a 15% equiaxial areal compression as the pre-strain is released. Then this device is used to impose compression to epithelial cells, and they observed a decrease in cell cross-sectional area and an increase in cell layer height. In my opinion, the authors’ effort is extensive. However I still have some comments as listed below.

1. Why the equiaxial stress is important to cells? Compared with the 2D planar stress, the cells fate would be different with the equiaxial stress? It would be much easier to impose 2D planar stress to cells. Is it really necessary to impose the equiaxial stress?

2. In figures 3E and 3F, the Z-displacement decreases first and then increases. What is the reason?

3. Sometimes, periodic stress is needed for cells. I am wondering if this device enables the periodic stress. If so, how many cycles can the membrane bear.

4. After culturing cells and adding the medium, will the mechanical properties (e.g., stress-strain curve) of the membrane change?

5. Figure 4D and Figure 5A are blurred, can the authors replace them with the colorful images.

Reviewer #2: Report on "Pneumatic equiaxial compression device ..." by Peussa et al.

The authors report on a pneumatic cell stretcher that can be operated also for lateral cell compression. To this end, they introduced a pressure reservoir that would hold the suction pressure during the short period of time needed to transport the device from the cell culture incubator to the microscope. Moreover, they optimized the design of the silicone rubber device to minimize vertical motion during radial compression.

After reading this manuscript and earlier work by the group, I find this manuscript describes incremental progress limited to technical aspects. The cell experiments performed with the improved instrument are of preliminary nature and reproduce well-established aspects of mechanobiology. The advantages of the set-up can be easily reproduced by mechanical devices based on translation stages. In those, you simply need to add a clamp or a locking screw to maintain a stable prestretch of the sample during cell culture and transport to the microscope. Moreover, a focus drift of about 40 to 50 microns as reported in the manuscript is good but not a major breakthrough. Overall, the material described here is fitting for the Supporting Information or the Materials and Methods part of a scientific paper but on its own it is not strong enough to justify a publication.

Besides this general and strong concern, I find the manuscript prepared in a less than careful way.

1) It is not clear at all if the silicone rubber membrane was bought (if so, where?) or manufactured in house (if so, how?).

2) The claim of high resolution in light microscopy is exaggerated. A reduction of the optical resolution by a factor of 2 (see Table Fig. 3G) is severe.

3) The authors should state which type of Silpuran they used. At this moment in time, Wacker is offering at least eleven different RTV silicones of the Silpuran series.

4) Using the acronym PDMS instead of giving the proper type (Sylgard 184) is lab jargon.

5) In the prototype experiment, Fig 4, an area compression of 15% was applied via the substrate contraction. However, the average cell area decreased by 30% and the thickness did more than double with time. This should be explained.

6) The distribution of individual measurement points (dots) in Fig 5B clearly shows a non-Gaussian distribution. Even though the t-test requires data that are normally (i.e. Gaussian) distributed, the authors used this statistical test. Therefore, the conclusions about significance are void.

Please note, this list is by no means exhaustive.

Reviewer #3: Review of PONE-D-22-04502

This paper is impressive in the details provided for the testing and evaluation of a 2-D compression device for manipulating cells grown as a monolayer or thin layers of cells on a substrate. It provides preliminary data on experiments carried out on cells and characterizes a variety of different substrate materials in terms of autofluorescence for use with the stretcher. Additional studies are included to analyze the variability due to manufacturing multiple wells. This is a valuable addition to an area that has received quite a bit of interest. Experimentalists have been developing uniaxial and biaxial ways of stretching membranes to provide uniform tensile strain fields many of which mentioned in the paper. While the idea proposed to pre-stretch a membrane to provide compression by releasing the pre-stretch, it may not be new it may be the first publication of such a device and a demonstration of its use.

The paper is well written, and the work is placed in context with prior studies. There are a few areas where I would suggest some minor changes to add further details, but other than that I see little to complain about and find this to be a nicely done and carefully caried out study.

1. Line 137 and 158 and in other places: I believe the word ‘stabilator’ should be replaced with ‘stabilizer’ if I understand its function correctly as a rigid barrier to prevent the outer wall from deforming.

2. I found it difficult to understand from Fig. 1 what the stabilizer is and does until I saw figure S1 in the Supplementary material. I suggest referencing this when first introducing the stabilizer.

3. The information on autofluorescence of the membranes is quite valuable for applications of these devices. Were you able to determine if here was any strain-dependence in the autofluorescence?

4. Line 213: The equation would be more useful to the general reader if the terms within the equation were defined, even if the calculation is a simple one.

5. Line 352: I am not sure I would characterize the strain response as being relatively linear with pressure based on the plots in Fig. 3. You state that hysteresis does not occur. How was that determined (over what length of time) and was that the case for each of the membrane materials? If pre-stretch was applied for 6 days in an incubator, as mentioned elsewhere, were you able to confirm that there were no strain changes due to stress relaxation at the temperature and humidity levels over that length of time?

6. You mention that the device applies equibiaxial strain. I assume from the description that the strain field is also uniform even close to the edges, although deformation of the inner wall may introduce disturbances in that. It might be worth mentioning the device also provides a uniform strain field.

7. One of the issues with devices like this is the fluid motion which can stimulate cells as well especially under cyclic loading. Have you been able to determine what the fluid effects are under cyclic loading when the side walls deform, since this would be likely to introduce some vertical motion of the fluid and fluid shear near the wall?

8. I was not able to find any related files in the https://zenodo.org/ database by searching using the article title.

6. PLOS authors have the option to publish the peer review history of their article (what does this mean?). If published, this will include your full peer review and any attached files.

Reviewer #1: No

Reviewer #2: No

Reviewer #3: **Yes: **John L. Williams

---

## [Author Response · Author response to Decision Letter 0]

21 Apr 2022

Response to Reviewers

Reviewer #1

In this paper, the authors designed an equiaxial cell compression device which is suitable for microscopy. The device can produce a maximum 8.5% radial pre-strain which leads to a 15% equiaxial areal compression as the pre-strain is released. Then this device is used to impose compression to epithelial cells, and they observed a decrease in cell cross-sectional area and an increase in cell layer height. In my opinion, the authors’ effort is extensive. However, I still have some comments as listed below.

We are grateful to the reviewer for carefully reading the manuscript and providing constructive criticism. Below we will address points raised by the reviewer.

1. Why the equiaxial stress is important to cells? Compared with the 2D planar stress, the cells fate would be different with the equiaxial stress? It would be much easier to impose 2D planar stress to cells. Is it really necessary to impose the equiaxial stress?

We are not absolutely sure what the reviewer means by 2D planar stress. Nevertheless, cells respond differently depending on the directions of the strain. The direction affects, for example, the migration and orientation of cells [1,2]. Therefore, devices for applying both kinds of strains (unidirectional and equiaxial) are required. The approach in this study focused on equiaxial strain. If the reviewer meant a non-symmetrical strain field, then again, cell response might be different. However, from the experimental point of view having a non-symmetrical strain field makes interpretation of results difficult as each cell experiences a different stimulation. A remark was added on lines 62 – 64.

2. In figures 3E and 3F, the Z-displacement decreases first and then increases. What is the reason?

The hinge structure affects the profile of the z-displacement. Because the upper part of the inner wall of the vacuum chamber is in a small angle and because the vacuum pressure produces pulling force perpendicular to the wall surface the net force targeting to the entire wall is slightly downwards at the beginning. Also, the inner corner (as shown in Fig. 1D) of the wall pushes the silicone membrane downwards thus compensating for the z-directional rising. After applying more vacuum pressure the pulling force of the thin membrane in the vacuum chamber starts to take a role and pull everything upwards. In other words, in the vacuum the angled wall structure compensates the upwards directed pulling effect of the membrane inside the vacuum chamber. This simulation study was also published in the COMSOL conference proceedings [3]. An explanation was added to lines 399 – 400.

3. Sometimes, periodic stress is needed for cells. I am wondering if this device enables the periodic stress. If so, how many cycles can the membrane bear.

The previous version of the device has been used to study the effects of cyclic equiaxial strain [4,5] on cardiomyocytes and adipose stem cells and cyclic unidirectional strain [2] on cardiomyocytes. The number of cycles that the device can bear has not been determined, but in these studies the continuous cyclic strain was successfully maintained up to 10 days and cells cultured up to 21 days in the device. The device presented in this manuscript is based on the same mode of action and on a similar pressure control system and therefore, could also be used for cyclic strain. This was implied in the introduction on lines 113 – 114. 

4. After culturing cells and adding the medium, will the mechanical properties (e.g., stress-strain curve) of the membrane change?

A stress-strain measurement was performed to a device containing media. As the plot from this liquid containing measurement falls under the normal variation seen in strain measurements, we conclude that the effect of media is irrelevant. However, this test was not performed in a scale large enough to determine reliable significance values. The data was added to S1 Fig to part D and mentioned on lines 395 – 396 and 774 – 776. Unfortunately, we do not have any systematic study of how cells affect the mechanical properties of the membrane. That is an interesting question though and we consider this as a future task to be performed.

5. Figure 4D and Figure 5A are blurred, can the authors replace them with the colorful images.

It is standard procedure to present confocal images in greyscale when channels are presented separately. Adding colors will not affect the resolution of the images. We modified Fig 4D by inverting the lookup table. This distinguishes the cell boarders better. However, the image is a projection from the orthogonal view of the stack. Resolution in confocal images is poorer in z-direction, which explains the blurriness. In Fig 5A a lower magnification objective (Plan Apo VC 20x DIC) was used in order to image large fields of view. This explains the poorer resolution in the images compared to Fig 4A (63x/1.20 W Korr M27). We also suspect that in the PDF file the images had been packed, and this had decreased the resolution. In the submitted TIFF-files image quality was better.

 

Reviewer #2 

Report on "Pneumatic equiaxial compression device ..." by Peussa et al. The authors report on a pneumatic cell stretcher that can be operated also for lateral cell compression. To this end, they introduced a pressure reservoir that would hold the suction pressure during the short period of time needed to transport the device from the cell culture incubator to the microscope. Moreover, they optimized the design of the silicone rubber device to minimize vertical motion during radial compression.

After reading this manuscript and earlier work by the group, I find this manuscript describes incremental progress limited to technical aspects. The cell experiments performed with the improved instrument are of preliminary nature and reproduce well-established aspects of mechanobiology. The advantages of the set-up can be easily reproduced by mechanical devices based on translation stages. In those, you simply need to add a clamp or a locking screw to maintain a stable pre-stretch of the sample during cell culture and transport to the microscope. Moreover, a focus drift of about 40 to 50 microns as reported in the manuscript is good but not a major breakthrough. Overall, the material described here is fitting for the Supporting Information or the Materials and Methods part of a scientific paper but on its own it is not strong enough to justify a publication.

Besides this general and strong concern, I find the manuscript prepared in a less than careful way.

We thank the reviewer for reading the manuscript and even prior papers from the group.

We would like to remind the reviewer about the Plos One editorial policy, which states “We evaluate submitted manuscripts on the basis of methodological rigor and high ethical standards, regardless of perceived novelty.” In addition, we find interesting that other 2 reviewers seem to have opposite opinion regarding the suitability of the manuscript for publication.

1) It is not clear at all if the silicone rubber membrane was bought (if so, where?) or manufactured in house (if so, how?).

The SILPURAN® silicone membrane used in the device was bought from Wacker Chemie Ltd. This is stated in the manuscript on lines 134. We have now specified on line 134 that the only film in the SILPURAN® product family, SILPURAN® FILM 2030, was used.

For comparison of autofluorescence and resolution, also SILPURAN® FILM 2030 in 100 µm thickness, ELASTOSIL® FILM 2030 in the thickness of 100 µm and 200 µm, and Gloss film (Special Manufacturing Inc.) in the thicknesses of 125 µm and 250 µm thicknesses were used. These are also commercial products, which is stated on lines 229 and 230. Also, an in-house manufactured film based on SYLGARD™ 184 (Dow) was used, and the manufacture process is presented in the supplementary information S1 Appendix. Corrections were also made to S1 Appendix on lines 30 and31.

2) The claim of high resolution in light microscopy is exaggerated. A reduction of the optical resolution by a factor of 2 (see Table Fig. 3G) is severe.

Here we disagree with the reviewer. The device allows the usage of high numerical aperture (NA > 1) objectives, which also often have short working distances. When optical imaging is considered then diffraction limited resolution can easily vary a lot, e.g., 10x/0.2 objective gives theoretical axial resolution of approx. 25µm, whereas 63x/1.2 WI objective yields approx. 0.7µm theoretical axial resolution. Thus, even when used membrane leads into reduced resolution when using 63x/1.2 WI objective, the resolution still is substantially high, allowing for example optical sectioning with a laser scanning confocal microscope. Thus, we feel that the term high resolution is justified. 

On lines 117 – 120 we state that the possibility for high resolution microscopy is based on the short working distance allowed by the 200 µm membrane thickness, not purely the membrane optical properties. As stated on line 416, the FWHM test was performed to optimize this high-resolution imaging feature. Furthermore, the reduction in resolution occurs mainly in z-direction, whereas no effect is seen in the xy-direction. We added this specification on lines 425 – 426:” However, no difference was seen in the in-plane (xy-directional) resolutions”.

3) The authors should state which type of Silpuran they used. At this moment in time, Wacker is offering at least eleven different RTV silicones of the Silpuran series.

See the answer to Question 1.

4) Using the acronym PDMS instead of giving the proper type (Sylgard 184) is lab jargon.

The reviewer is correct that PDMS is a broad term and all silicone membranes used in this study are PDMS-based. However, the most well-known PDMS type, used in similar studies in the field, is Sylgard 184 (Dow) and it is called “PDMS” in over 10 000 articles listed in PubMed. Therefore, in this manuscript we have defined and specified the abbreviation PDMS to correspond specifically to Sylgard 184 (Dow). We state the more accurate description of the used PDMS in the Materials and Methods section on line 175. 

5) In the prototype experiment, Fig 4, an area compression of 15% was applied via the substrate contraction. However, the average cell area decreased by 30% and the thickness did more than double with time. This should be explained.

We thank the reviewer for this observation. We want to point out that directly after compression, when the purely mechanobiological effect is visible, the decrease in area is 21%. In our opinion this is reasonably close to the 15% areal compression produced by the device. The slower effects that can be seen during the following 4 hours might be more biological responses and not necessarily directly related to the 15% areal compression. Additionally, we suspect that this interesting phenomenon is partly caused by the fact that actomyosin contraction as well as occludin, which was used for determining the cell cross-sectional areas, are apically located [6]. Therefore, cross-sectional cell areas were determined from the part of the cell that undergoes most contraction. The decrease in cell cross-sectional area can partly be explained also by the large variation in cell sizes as evident in the standard deviation presented in the figure. In the Fig 4C, data is pooled from 4 different experiments, and during each experiment the different areas were imaged at different time points. This increases the variation and explains why decrease is higher than theoretically expected. We modified Fig4C to show the mean decrease (%) in cell area instead of the area (µm2), as we considered it to be more informative.

The same applies for the increase in cell layer thickness. The purely mechanical response directly after compression produced a 20% increase in thickness, which concurs with the decrease in cross-sectional area. Again, the slower responses can be accounted for by biological responses, for example osmosis. Finally, variation in measurement data might affect the averages. For this reason, the standard deviation is presented. 

Explanation was added on lines 463 – 466 and 474 – 746 and discussed on lines 595 – 612.

6) The distribution of individual measurement points (dots) in Fig 5B clearly shows a non-Gaussian distribution. Even though the t-test requires data that are normally (i.e. Gaussian) distributed, the authors used this statistical test. Therefore, the conclusions about significance are void.

We thank the reviewer for the statistical expertise. A nonparametric significance test was performed to account for the non-normality of the data’s distribution. According to independent samples Mann-Whitney U test the significance is still high, p<0,0001. Corrections were made on lines 324, 495 – 497 and 504.

Please note, this list is by no means exhaustive.

We note these constructive criticisms by the reviewer. 

 

Reviewer #3

Review of PONE-D-22-04502. This paper is impressive in the details provided for the testing and evaluation of a 2-D compression device for manipulating cells grown as a monolayer or thin layers of cells on a substrate. It provides preliminary data on experiments carried out on cells and characterizes a variety of different substrate materials in terms of autofluorescence for use with the stretcher. Additional studies are included to analyze the variability due to manufacturing multiple wells. This is a valuable addition to an area that has received quite a bit of interest. Experimentalists have been developing uniaxial and biaxial ways of stretching membranes to provide uniform tensile strain fields many of which mentioned in the paper. While the idea proposed to pre-stretch a membrane to provide compression by releasing the pre-stretch, it may not be new it may be the first publication of such a device and a demonstration of its use. The paper is well written, and the work is placed in context with prior studies. There are a few areas where I would suggest some minor changes to add further details, but other than that I see little to complain about and find this to be a nicely done and carefully caried out study.

We thank the reviewer for the positive and constructive review and feedback. We will address the reviewer comments below. 

1. Line 137 and 158 and in other places: I believe the word ‘stabilator’ should be replaced with ‘stabilizer’ if I understand its function correctly as a rigid barrier to prevent the outer wall from deforming.

 This term has been corrected in the manuscript.

2. I found it difficult to understand from Fig. 1 what the stabilizer is and does until I saw figure S1 in the Supplementary material. I suggest referencing this when first introducing the stabilizer.

A reference to S1 Fig has been added on line 140 where the stabilizer ring is first introduced.

3. The information on autofluorescence of the membranes is quite valuable for applications of these devices. Were you able to determine if here was any strain-dependence in the autofluorescence?

Unfortunately, this was not tested in the autofluorescence measurements. However, this is an interesting question, and the authors will consider determining this in the future.

4. Line 213: The equation would be more useful to the general reader if the terms within the equation were defined, even if the calculation is a simple one.

 The terms used in the equation are now defined on lines 215 – 221.

5. Line 352: I am not sure I would characterize the strain response as being relatively linear with pressure based on the plots in Fig. 3. You state that hysteresis does not occur. How was that determined (over what length of time) and was that the case for each of the membrane materials? If pre-stretch was applied for 6 days in an incubator, as mentioned elsewhere, were you able to confirm that there were no strain changes due to stress relaxation at the temperature and humidity levels over that length of time?

The description of the stress-strain behavior was corrected to:” Within this range, strain is relatively linear up to ~-200 mbar (linear fit R2=0.98) but continues to increase up to -350 mbar” on lines 358 – 360.

Hysteresis was measured only for the used silicone membrane, SILPURAN® 200 µm. It was determined by performing strain measurements at 20 mbar intervals both as the pressure was decreased (membrane stretched) and as pressure was returned back to relaxation (compression). Data were plotted in the same graph (in Fig 3A and 3B, measurements form the center and from the edge of the device, respectively), and as seen, data points overlap suggesting that pressures produce equal strains in loading and unloading. These stretch-compression measurements took about 1 hour. This was specified on line 362 – 363: “Strain was measured both as negative pressure was increased from -4 mbar to -350 mbar (line called stretch in Fig3A and 3B), and as pressure was returned to the initial -4 mbar (line called compression in Fig3A and 3B). These lines concur, demonstrating that hysteresis does not occur at least within this strain range. Furthermore, the strains are similar both in the center of the device (Fig3A) and in the edges and center of the device (Fig 3B) suggesting that the device creates a uniform strain field.”

As PDMS-based materials are known to have low hysteresis [7], we did not conduct long term experiments. Instead, we verified that the vacuum battery is able to maintain a steady pressure during transportation (S1 Appendix). 

We have not characterized the effect of temperature and humidity to the SILPURAN® membrane. 

6. You mention that the device applies equibiaxial strain. I assume from the description that the strain field is also uniform even close to the edges, although deformation of the inner wall may introduce disturbances in that. It might be worth mentioning the device also provides a uniform strain field.

Previously, we attempted to measure entire strain fields. This method requires imaging a large field of view and thus forces to use smaller magnification. This caused the displacements of analyzed particles to be very small (movement of only few pixels). For this reason, we came to the conclusion that measuring the strain field was very inaccurate and prone to error. Therefore, we instead performed detailed measurement in higher magnification at 4 different locations throughout the stretching area. This same measurement method was used also in our previous publication [4]. In these more accurate measurements, we saw similar stress-strain behavior in the center of the device and at various points near the edge. This was elaborated on lines 363 – 364:” Furthermore, the strains are similar both in the center of the device (Fig3A) and in the edges and center of the device (Fig 3B) suggesting that the device creates a uniform strain field.”

7. One of the issues with devices like this is the fluid motion which can stimulate cells as well especially under cyclic loading. Have you been able to determine what the fluid effects are under cyclic loading when the side walls deform, since this would be likely to introduce some vertical motion of the fluid and fluid shear near the wall?

Thank you for bringing up this subject. We are aware that shear stress can affect cell fate and thus reducing shear stress was the one design parameter when this structure was initially designed. Even though cyclic stretching was not the topic in this manuscript we have, indeed, studied (computer simulations) the effect of shear stress and explained that in our previous paper [2]. Although this computational simulation was previously done for unidirectional stretching it demonstrates, in our opinion, that with the current design the shear stress does exists, but it is negligible, being < 1.5mPa. The shear stress in our device is far less than for example in the commercial Flexcell® device where the up-and-down movement of the stretching membrane inside the culture chamber creates culture medium circulation and therefore also relatively large shear stress (132mPa) for the cells [2,8].

8. I was not able to find any related files in the https://zenodo.org/ database by searching using the article title.

According to PlosONE instructions it is not required to share the raw data until the manuscript is accepted. The raw data will be added to the supplied database upon acceptance. 

List of references:

1. Pennisi C, Olesen C, de Zee M, Rasmussen J, Zachar V. Uniaxial Cyclic Strain Drives Assembly and Differentiation of Skeletal Myocytes. Tissue Eng Part A. 2011;17: 2543–2550. doi:10.1089/ten.TEA.2011.0089

2. Kreutzer J, Viehrig M, Pölönen R-P, Zhao F, Ojala M, Aalto-Setälä K, et al. Pneumatic unidirectional cell stretching device for mechanobiological studies of cardiomyocytes. Biomech Model Mechanobiol. 2019. doi:10.1007/s10237-019-01211-8

3. Mäki A-J, Kreutzer J, Kallio P. Optimizing Elastomeric Mechanical Cell Stretching Device. COMSOL Conference 2018 Lausanne. 2018. Available: https://www.comsol.com/paper/optimizing-elastomeric-mechanical-cell-stretching-device-64511

4. Kreutzer J, Ikonen L, Hirvonen J, Pekkanen-Mattila M, Aalto-Setälä K, Kallio P. Pneumatic cell stretching system for cardiac differentiation and culture. Med Eng Phys. 2014;36: 496–501. doi:https://doi.org/10.1016/j.medengphy.2013.09.008

5. Virjula S, Zhao F, Leivo J, Vanhatupa S, Kreutzer J, Vaughan TJ, et al. The effect of equiaxial stretching on the osteogenic differentiation and mechanical properties of human adipose stem cells. J Mech Behav Biomed Mater. 2017;72: 38–48. doi:https://doi.org/10.1016/j.jmbbm.2017.04.016

6. Pearl EJ, Li J, Green JBA. Cellular systems for epithelial invagination. Philos Trans R Soc B Biol Sci. 2017;372: 20150526. doi:10.1098/rstb.2015.0526

7. Zhengjin W, Chunping X, Xi Y, Paul LF, Julien M, Zhigang S. Stretchable materials of high toughness and low hysteresis. Proc Natl Acad Sci. 2019;116: 5967–5972. doi:10.1073/pnas.1821420116

8. Thompson MS, Abercrombie SR, Ott C-E, Bieler FH, Duda GN, Ventikos Y. Quantification and significance of fluid shear stress field in biaxial cell stretching device. Biomech Model Mechanobiol. 2011;10: 559–564. doi:10.1007/s10237-010-0255-1

---

## [Decision Letter · Decision Letter 1]

3 May 2022

Pneumatic equiaxial compression device for mechanical manipulation of epithelial cell packing and physiology

PONE-D-22-04502R1

Dear Dr. Ihalainen,

We’re pleased to inform you that your manuscript has been judged scientifically suitable for publication and will be formally accepted for publication once it meets all outstanding technical requirements.

Kind regards,

Antonio Riveiro Rodríguez, PhD

Academic Editor

PLOS ONE

Reviewers' comments:

Reviewer's Responses to Questions

**Comments to the Author**

1. If the authors have adequately addressed your comments raised in a previous round of review and you feel that this manuscript is now acceptable for publication, you may indicate that here to bypass the “Comments to the Author” section, enter your conflict of interest statement in the “Confidential to Editor” section, and submit your "Accept" recommendation.

Reviewer #1: All comments have been addressed

Reviewer #3: All comments have been addressed

2. Is the manuscript technically sound, and do the data support the conclusions?

Reviewer #1: Yes

Reviewer #3: Yes

3. Has the statistical analysis been performed appropriately and rigorously? 

Reviewer #1: (No Response)

Reviewer #3: Yes

4. Have the authors made all data underlying the findings in their manuscript fully available?

Reviewer #1: (No Response)

Reviewer #3: Yes

5. Is the manuscript presented in an intelligible fashion and written in standard English?

Reviewer #1: (No Response)

Reviewer #3: Yes

6. Review Comments to the Author

Reviewer #1: (No Response)

Reviewer #3: My questions and concerns have all been addressed. I assume that the database posted in zenodo.org will be made publicly available after the paper is accepted.

7. PLOS authors have the option to publish the peer review history of their article (what does this mean?). If published, this will include your full peer review and any attached files.

Reviewer #1: No

Reviewer #3: **Yes: **John L. Williams

---

## [Editor Report · Acceptance letter]

24 May 2022

PONE-D-22-04502R1 

Pneumatic equiaxial compression device for mechanical manipulation of epithelial cell packing and physiology 

Dear Dr. Ihalainen:

I'm pleased to inform you that your manuscript has been deemed suitable for publication in PLOS ONE. Congratulations! Your manuscript is now with our production department. 

Kind regards, 

on behalf of

Dr. Antonio Riveiro Rodríguez 

Academic Editor

PLOS ONE